# Ideas: a simple proposal to improve the contribution of IPCC WG1 to the assessment and communication of climate change risks

**Rowan T. Sutton**

National Centre for Atmospheric Science, Department of Meteorology, University of Reading, RG6 6BB, UK

*Correspondence to*: Rowan Sutton (rowan.sutton@ncas.ac.uk)

**Abstract:** The purpose of the Intergovernmental Panel on Climate Change (IPCC) is to provide policy relevant assessments of the scientific evidence about climate change. Policy making necessarily involves risk assessments, so it is important that IPCC reports are designed accordingly. This paper proposes a specific idea, illustrated with examples, to improve the contribution of IPCC Working Group I to informing climate risk assessments.

The process of drafting the Working Group I (WGI) contribution to the Sixth Assessment Report of the Intergovernmental Panel on Climate Change (IPCC AR6) began recently with the first Lead Author meeting held in Guangzhou, China, in June 2018. An issue that merits greater attention than in previous WGI reports is the assessment and communication of risk. It is now widely accepted that it is appropriate - and necessary for decision making - to frame climate change as a problem in risk
assessment and risk management (King et al, 2015; Weaver et al 2017). In the AR5 greater use was made than in previous Assessment Reports of a formal risk assessment framework which spans the dimensions of hazard, exposure and vulnerability (IPCC, 2014). However, risk framing had little influence on the WGI report, and this should be addressed in AR6.

A common measure of risk is likelihood x impact (Fig 1). It is standard practice in risk assessment to highlight both the most likely impacts *and* low likelihood high impact scenarios. Such scenarios merit specific attention because the associated costs can be extremely high, so decision makers need to know about them. It follows that WGI has a responsibility to assess and communicate explicitly the scientific evidence concerning potential high impact scenarios, even when the likelihood of
occurrence is assessed to be small. In past reports the assessment of key parameters by WG1 has focussed overwhelmingly on likely ranges only. When information has been provided about the tails of distributions only likelihoods have been communicated using terms - following the IPCC's uncertainty guidance (Mastrandrea et al, 2010) - such as "very unlikely" or "extremely unlikely": a clear steer that policy makers should largely ignore such possibilities. But this is wrong. Policy makers care about risk not likelihood alone. The IPCC's uncertainty guidance is valuable, but by itself it is insufficient to
guide the assessment of risk. In particular, the focus on likelihood terminology that is symmetric with respect to high and low impact scenarios downplays the importance of low likelihood high impact risks (Fig 1).

I suggest the WGI authors should agree a modest number of key parameters for which an assessed ***physically plausible high impact scenario (PPHIS) or storyline*** (e.g. Zappa and Shepherd, 2017) can be provided. This should be done for core parameters such as climate sensitivity and TCRE (the Transient Climate Response to cumulative carbon Emissions: Allen et al, 2009; Matthews et al, 2009), and could also be done for some large-scale impact-relevant metrics (informed by WGII), such as the magnitude of increases in extreme rainfall. There will be a need to agree consistent procedures for the definition, description, and use of such storylines, for example they could be associated with a specific assessed likelihood, and their characterisation should emphasise physical constraints and evidence, not model results alone. This will be helped by a growing literature on which to draw (e.g. Hazeleger et al, 2015; Zappa and Shepherd, 2017). Physically based high impact storylines are distinct from socioeconomic scenarios, but the WGI report could usefully provide information on outcomes that could arise from a combination of, e.g., high climate sensitivity and a high emissions scenario.

## Practical Implementation and Examples

WGI could adopt a practical definition of Physically Plausible High Impact Scenarios (PPHIS) along the following lines:

> An assessed physically based storyline for specific aspects of future climate change that is consistent with all available evidence and would result in impacts that are substantially greater than those implied by the relevant *likely* range.

The characterisation of each PPHIS should include: 1) an assessment of likelihood, and 2) an assessment of impact explicitly framed in conditional terms (i.e. conditional on the PPHIS being realised in the real world), with separate assessed confidence levels for each of these two components. This approach is in line with the IPCC uncertainty guidelines (Mastrandrea et al, 2010) which state: "For findings (effects) that are conditional on other findings (causes) … [author teams should] consider independently evaluating the degrees of certainty in both causes and effects, with the understanding that the degree of certainty in the causes may be low".

With regard to likelihood, I propose that WG1 should base PPHIS on scenarios that are assessed to be *very unlikely* (0-10%) rather than *extremely unlikely* (0-5%) or *exceptionally unlikely* (0-1%). In the context of deep uncertainty attempts to quantify the likelihood of a PPHIS more precisely are unlikely to be fruitful, and are not necessary to provide information that is useful for risk assessment (see e.g. www.deepuncertainty.org). Information about impacts should be limited in WGI to physical climate variables but should be quantitative where possible and include an assessed confidence level. WGII could make use of the WGI PPHIS to provide further information about impacts; this would help coordination between the working group reports and the production of the Synthesis Report.

Potential abrupt changes have long been recognised as an important risk-relevant issue for IPCC WG1 to assess (e.g. Section 12.5.5 in Collins et al, 2013). However, abrupt changes are only a subset - and not obviously the most important subset - of

PPHIS. It is notable that hardly any information about abrupt changes was included in the AR5 WGI Summary for Policymakers, and where information was included (e.g. for the AMOC, Section E.4 in IPCC, 2013), it addressed likelihood only with little or no information provided about impact.

Below are three examples of how PPHIS could be used by WGI, adapted from the WGI AR5 Summary for Policymakers. In these examples all the information used can be found somewhere within the AR5 report, but the synthesis and communication (including framing) of this information is different.

1. ECS

It is *very unlikely* that ECS is greater than 6$^{o}$C (*medium confidence*) but this value may be considered a Physically Plausible High Impact Scenario (PPHIS). If realised, such a value for ECS would *very likely* result in an increase in global mean temperature by 2100 well above 2$^{o}$C relative to 1850-1900 under all RCP scenarios except RCP2.6 (*high confidence*).

2. Sea level

A partial collapse of the marine-based sectors of the Antarctic ice sheet is considered *unlikely* during the 21$^{st}$ century (*medium confidence*). However, if realised this PPHIS could cause an additional contribution to sea level rise of up to several tenths of a meter during the 21$^{st}$ century (*medium confidence*).

3. Atlantic Meridional Overturning Circulation (AMOC)

It is *very unlikely* that the AMOC will undergo an abrupt transition or collapse in the 21$^{st}$ century for the scenarios considered (*medium confidence*). However, if it did occur such a transition would have very large rapid (decadal timescale) impacts on the regional climate of the North Atlantic and surrounding continents (*high confidence*) and substantial impacts on the climate of regions further afield (*medium confidence*). [More quantitative information on impacts could and should be provided.]

## Concluding remarks

Some will argue that the WGII report is needed to provide information on impacts. For detailed information this is certainly the case, but the general shape of the damage function for a large basket of impacts (Fig 1) is insensitive to such details, and is all that is needed to justify WGI providing a much more thorough assessment of relevant scenarios. Other critics will

suggest that for WGI to identify high impact scenarios explicitly would constitute scaremongering; this concern is no doubt one reason why previous WGI reports have focused so much on the likely range. But it is misguided. (See also Emanuel, 2014.) Policy makers need to know about high impact scenarios and WGI has a responsibility to contribute its considerable expertise to making the appropriate assessments.

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

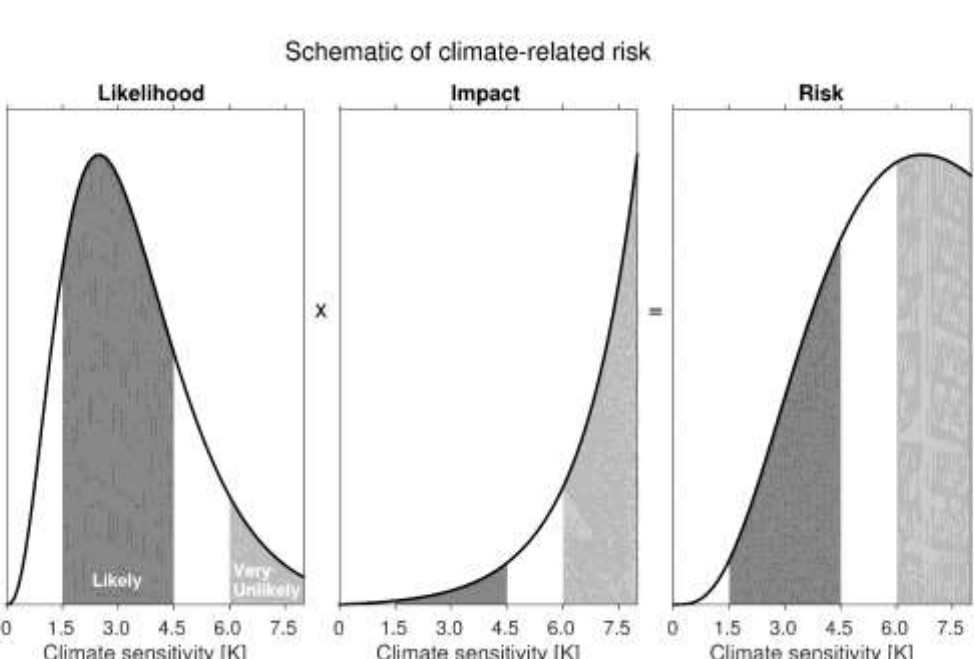

**Figure 1: A** *schematic* **representation of how climate change risk depends on equilibrium climate sensitivity (ECS).** Panel a shows a possible likelihood distribution consistent with the IPCC AR5 assessment that "Equilibrium climate sensitivity is likely in the range 1.5 C to 4.5 C (high confidence), extremely unlikely less than 1 C (high confidence) and very unlikely greater than 6 C (medium confidence)." Panel b illustrates schematically the fact that the cost of impacts and adaptation rises very rapidly (shown here as an exponential damage function) with ECS. Panel c shows that, in this example, the resultant risk (quantified here as likelihood x impact) is highest for high ECS values. The precise shape of the risk curve is dependent on assumptions about the shape of the likelihood and damage functions at high sensitivity (Weitzman et al, 2011). (Figure by Ed Hawkins.)

**Acknowledgments**

I would like to thank Ed Hawkins for making the figure, and for valuable discussions. I would also like to thank Ted Shepherd and all the reviewers and referees for their valuable comments, which improved the paper.