# Peer review of "Ideas: a simple proposal to improve the contribution of IPCC WG1 to the assessment and communication of climate change risks"

_Earth System Dynamics, 2018_

## Referee Comment (RC1) · S. Hallegatte (Referee) · 12 Jun 2018

I welcome the call to climate scientists to explore plausible high-impact scenarios, even if the likelihood of these scenarios is considered relatively low. Climate policy is about risk management, and risk management has always had to consider relatively extreme scenarios, with high impacts and low likelihood.

I would suggest however not to define those scenarios by their likelihood. Indeed, estimating the likelihood of non-likely scenario is difficult and controversial, and not necessary to provide useful inputs into risk assessment and the design of risk management policies and measures.

[Figure]

Stress testing in the financial sector, or the decision to insure a house again fire, usually does not involve an estimate of the probability of a given scenario. The idea is that testing a system against an extreme scenario provides useful insights into a system vulnerability, and help make the system stronger. "Robust decision-making" — a methodology that is becoming increasingly common in infrastructure design; see Lempert et al (2013) — and other methodologies to make decisions under deep uncertainty do not rely on the probabilities of every scenarios (even though most approaches can use probabilities when available). (See below for illustrations of how this approach is applied in various sectors, and www.deepuncertainty.org for more information.)

Constraining the proposed scenarios to those with an estimated probability may limit the range of possible futures that can be considered. And for stress testing, being exhaustive (i.e. considering all possible threats) is more important than being precise (i.e. quantifying the likelihood of a scenario and expected impacts).

The approach proposed here could be extremely valuable, and have direct operational implications. To ensure proper risk screening for long-term investments or planning, it is indeed critical to have an idea of the full range of possible futures. A great contribution from climate sciences would be to guide the selection of the most extreme scenarios that should be considered in such analyses, answering questions like "What is the range of plausible change in extreme rainfall in Europe by 2050 or by 2100?".

While the exact definition of these extreme scenarios will depend on the problem at stake (as one wants to be more pessimistic when catastrophic outcomes are possible), any guidance from climate scientists and models would be a great contribution to the design of more resilient infrastructure, and thus to a lower vulnerability to future climate change impacts. I think therefore that this manuscript should be published.

I would like to insist however on the communication challenge that the exploration of "extreme" scenarios creates. It would be useful if the authors — or some of his readers — could provide constructive ideas to ensure that IPCC readers understand the status

of unlikely scenarios and do not confuse them with forecasts or predictions.

Lempert, R.J., Popper, S.W., Groves, D.G., Kalra, N., Fischbach, J.R., Bankes, S.C., Bryant, B.P., Collins, M.T., Keller, K., Hackbarth, A., Dixon, L., LaTourrette, T., Reville, R.T., Hall, J.W., Mijere, C., McInerney, D.J., 2013. Making Good Decisions Without Predictions. Rand Corp. Res. Brief 9701.

Application of "decision-making under deep uncertainty" in various sectors:

Water: http://documents.worldbank.org/curated/en/617161468187788705/Robust-decision-making-in-the-water-sector-a-strategy-for-implementing-Lima-s-long-term-water-resources-master-plan

Energy: http://documents.worldbank.org/curated/en/465701468330278549/Making-informed-investment-decisions-in-an-uncertain-world-a-short-demonstration

Transport: http://documents.worldbank.org/curated/en/691821490628878185/Improving-the-resilience-of-Perus-road-network-to-climate-events

Hydropower: http://documents.worldbank.org/curated/en/179901476791918856/South-Asia-Investment-decision-making-in-hydropower-decision-tree-case-study-of-the-upper-Arun-hydropower-project-and-Koshi-basin-hydropower-development-in-Nepal

---

## Short Comment (SC1) · 12 Jun 2018

This is a well-intentioned call-to-arms but I wonder if it is misdirected? Yes policy makers need assessments of risk, coupled with assessments of vulnerability, but are IPCC WG1 scientists are best placed to provide this?

WG1 scientists and authors can assess the impacts on the physical climate system of high-end scenarios and sensitivities, abrupt changes etc. Certainly, more could certainly be done in this area as there are numerous gaps in the literature. However, providing a full assessment of physical risks, the risks to human and natural systems, the vulnerability of those human and natural systems to risks and assessments of

techniques to mitigate impacts, arguably requires input from all three IPCC working groups. (This is part of the rationale for the cross-working-group special reports which are currently being written.)

I think what the author might better argue for is greater working between scientists from the different working groups to produce more coordinated literature upon which such syntheses are made. And for community-coordinated projects such as CMIP to provide the infrastructure to underpin such syntheses.

The variable the author chooses to illustrate the argument raises an interesting side point. The AR5 assesses the equilibrium climate sensitivity as being extremely unlikely less than 1°C (high confidence), and very unlikely greater than 6°C (medium confidence). Yet, there are no models in the CMIP5 archive with such low and high climate sensitivities. Should modellers be seeking to produce models that sample more the tails of such distributions? Should MIPS be seeking to produce more extreme model response such as AMOC collapse or mega-tropical cyclones? Perhaps that is another discussion article.
* * *

---

## Short Comment (SC2) · 12 Jun 2018

Another way of framing this discussion is in the relative attention paid to avoiding Type 1 (false positive) and Type 2 (false negative) errors. I think it's fair to say that WG1 has emphasized the avoidance of Type 1 errors. Indeed many climate scientists would no doubt argue that this is the scientifically rigorous approach. However as Lloyd & Oreskes (2018 Earth's Future doi: 10.1002/2017EF000665) point out, any applied science has to consider both Type 1 and Type 2 errors, with the relative emphasis depending on the context. For example, in assessing the efficacy of a new drug, the tests guard against Type 1 errors, but in assessing possible detrimental side-effects,

the tests guard instead against Type 2 errors.

If the climate science community guards only against Type 1 errors, then it runs a large risk of making Type 2 errors. Such a conservative approach should not be equated with scientific rigour (Lloyd & Oreskes 2018). And it clearly compromises the ability to embrace the risk framework that IPCC is working towards.

How far the WGI report can go in this direction remains to be seen. It is not a simple matter, especially under the consensus framework within which IPCC operates. But if it fails to do so, then I fear that WGI will lose its relevance as it will have failed to address the most pressing questions currently facing climate science.
* * *

---

## Author Comment (AC1) · 19 Jun 2018

I thank Mat Collins for his comment on my proposal. His initial remarks reflect the statement at the beginning of my final paragraph: "Some will argue that the WGII report is needed to provide information on impacts." But as I argue in the same paragraph, very little information about impacts is needed to recognise that - for example - high climate sensitivity would lead to greater and more costly impacts than low climate sensitivity. And this is not controversial.

We agree that the cross-WG reports (and the underlying inter-disciplinary collaborations) have an important role to play in meeting the needs of policy makers, and as

such they are an important step forward. But in AR6 we also have a new cycle of the traditional three "siloed" Assessment Reports, so it is appropriate to ask how these reports can be improved – in particular how they can assess more effectively what policy makers need to know. I argue this requires greater attention by WGI to the explicit assessment of high impact scenarios, even when their likelihood is assessed to be low.

I agree the question of whether effort should be expended on developing models which more effectively sample high (or low) impact scenarios is an interesting one, and is largely beyond the scope of my proposal. However, the characterisation of Physically Plausible High Impact Scenarios (PPHIS) should be based on assessment of all relevant evidence and – as I state in the proposal – "not model results alone".

---

## Author Comment (AC2) · 19 Jun 2018

I thank S. Hallegate for his positive response to my proposal, his linking it to wider practice in risk assessment, and his very valuable suggestions. I particularly welcome the remark that "A great contribution from climate sciences would be to guide the selection of the most extreme scenarios that should be considered. . .", and wholeheartedly agree.

With regard to the point that high impact scenarios should not be defined by their likelihood, I agree that attempts to assess likelihood with inappropriate precision would be misplaced (particularly in the context of deep uncertainty). However, there is a need

to make connection with the IPCC calibrated language for confidence and likelihood. This is related to the communication challenge, which I also agree is extremely important. To address both points I offer the following more specific proposal, and I would welcome suggestions to further improve or refine it.

IPCC WG1 could define a Physically Plausible High Impact Scenario (PPHIS) as: an assessed physically-based storyline for specific aspects of future climate change that is consistent with all available evidence and would result in impacts that are significantly greater than those implied by the relevant likely range. It is proposed that where possible WG1 should base the assessment of PPHISs on scenarios that are assessed to be very unlikely (0-10%) rather than extremely unlikely (0-5%) or exceptionally unlikely (0-1%). Information about the impacts of these scenarios should be provided, but explicitly framed in conditional terms (i.e. conditional on the PPHIS being realised in the real world) and together with an assessed confidence level following the IPCC guidelines.

Such a definition should ensure no confusion between PPHISs and forecasts or predictions. It is worth noting that potential abrupt or irreversible changes (which were discussed in 12.5.5 of the WG1 contribution to AR5) are a subset (but only a subset) of PPHISs.

A specific example, adapted from the AR5 Summary for Policymakers, might result in an assessment such as: "It is very unlikely that ECS is greater than 6C but this value may be considered a Physically Plausible High Impact Scenario (PPHIS). If realised, such a value for ECS would very likely result in an increase in global mean surface temperature by 2100 well above 2C relative to 1850-1900 under all RCP scenarios except RCP2.6 (high confidence)."

---

## Short Comment (SC3) · 20 Jun 2018

The comments relate to the meteorological significance of the high impact events which occur at the tail of the spectrum. For example, the 1607 (Julian calendar) flood or the 1703 cat 2 hurricane which destroyed much of the British navy at Chatham. These rare, but high impact events are described by process that cannot be "permitted" or climate models. Gadian et al (2018), doi: 10.1002/joc.5336 demonstrated that convectively permitting solutions capture more than 10 times the number of convectively param- eterised results for a control period of (1989-1995) and further form a same period (2030-2036) underestimate their importance by a further 20%. This involves compar-

ing simulations with convective eparameteriatsion of < 12km resolution and convectively permitting solutions of < 3km resolution. This is mirrored on other processes such as longer dry spell durations etc.

Thus, I would go further than Kerry Emmanual's comment (10th June 2018) which relates to the professionalism of the "meteorologist". It is more than a "strong professional obligation to estimate an portray the entire probability distribution ...but also the high end risk function , because of the outcome function is very high there". It is a moral issue required of meteorologists, who know they cannot model the extreme processes, not to emphasise and put effort into including or even primitively estimating the low frequency tail impacts. I agree that "policy makers need to know has a responsibility to know", but also also the climate modellers / meteorologists have a moral responsibility to provide estimates for which those making the appropriate estimations of impact. It is the small scale processes which are often important, even in maintaining large structures such as blocking anticyclones, and it is similarly disingenuous the meteorologists not to bring the importance of the tail and the fact that the climate modellers are unable to represent.

Sutton argues that it is "misguided" that WG1 has focused in the likely range. However, I also agree with Emmanuel, that it is "misguided" the climate modellers themselves have focused so much on the likely range, and the bulk of the distribution rather than the extremes in the tails. For a multitude of reasons the climate scientists have almost deliberately ignored the tail because they cannot understand what is happening there. However, I would again go further that in IPCC AR6 , a group of core, (mainly climate) scientists has been selected who are experts on the climate models which represent the bulk of the spectrum, with only a few selected who understand and are experts in the smaller scaled meso / micro scale meteorological, thermodynamical physics. If only a few are present who understand the tail, then WG1 is very unlikely to represent the risks

The "Tail Risk vs Alarmism" is always an argument proposed by the more conservative

elements of the scientific establishments, and the peer review process where only the consensus is published and careers established. If the you are considering building a new fast breeder nuclear reactor on a coastal site, without adequate protection, because IPCC was too focused on the bulk of the spectrum, in an area where a flood in 1607 a meteorological event caused more than 10m of flooding and also where severe inundation has occurred in close proximately, then it is more than a matter of integrity. It is a moral act, for the meteorologists and WG1 to include these tail effects. Succumbing the charge being "alarmist" without making a stand, is an act of cowardice and dishonour, even if it does, and will, wreck ones career. Emmanual's comment is correct that it makes scientists "skittish", but it is also an understatement. Climate modellers, which I am afraid includes many in the UK, and especially those scientists in WG1 and those who selected them, have a huge responsibility which history will hold them liable for if one of those events occur.

In summary, Sutton's argument that it is basically ignoring the tail is misguided although correct is a gross understatement and is no where near self critical enough. The tail end processes so eminently described by Emmanual, are the ones that WG1 should be predominantly studying, but which I suggest will be unqualified to asses, and a direct consequence of the past decades of climate science in may national institutions , and the source of many careers.

---

## Short Comment (SC4) · 21 Jun 2018

The reference to my comments by Kerry Emmanual can be found:-

https://ourchangingclimate.wordpress.com/tag/kerry-emanuel/

Alan Gadian

---

## Short Comment (SC5) · 26 Jun 2018

I like aspects of this and agree that WG1 can do a lot better and supporting the risk framing than in past reports. I also think WG1 should not try and go it alone but take input from WGII authors about which risks they care about. I also think one part of the pdf should not be focussed on in preference to other parts - the. likely range and bottom tail are just as important.

Lastly, small biases and differences in assumptions really affect the tails of the distributions. Therefore, I would strongly advise against trying to quantify the shape of these too closely.

---

## Short Comment (SC6) · 3 Jul 2018

If the tails of the distribution are affected so much by assumptions, then who else but WG1 authors or their peers can interpret them correctly? Physically Plausible High Impact Scenarios (PPHIS) can be a valuable tool for WG1 to demonstrate through example to the wider community what the sensible and less sensible ways are of interpreting the tail probabilities. Moreover, by asking the community of physical climate scientists to be explicit about the assumptions, the work on PPHIS may feed back into an improved understanding of what the tail ends signify.

If not WG1 authors or their peers, it is hard to see who else could be doing the inter-

pretation. Decision making that is informed by a probabilistic measure of uncertainty sooner or later in the decision making process has to take one (or at most a few) point measurement(s) of the underlying probabilistic measure. This "collapse" of the uncertainty generates one (or a few) number(s) that are then interpreted almost deterministically downstream from the point measurement. This may not be a desirable situation, but it is an almost inevitable aspect of decision making in the real world. Several factors contribute to this. People tend to be poor intuitive statisticians - see (Sanborn and Chater, 2016) for a recent review and explanation in terms of brain operation. In absence of a formal training in probability, collapsing the uncertainty is therefore a natural and often the only way for people (including many decision makers) to make sense of a problem. Many decision making problems also call for yes/no answers, for a choice between a limited set of options, or for a determinate boundary for which to design a solution for. This means that collapsing the uncertainty is a necessary part of coming to actionable decisions.

The solution that the IPCC (and specifically WG1) has adopted for the uncertainty inherent in climate science is to define a formal methodology to report the scientific uncertainty, and to leave the sense making (i.e., the collapsing) to its downstream users. To some degree, this is required as the question "What do we care about with regards to this decision?" can only be answered in the realm of values and politics, not in the realm of science.

However, this situation has also led to very unfortunate consequences, as it leaves the collapsing of the uncertainty to downstream users, regardless of their understanding or skill to do so. There are many signs that this has led to a wrong focus or misinterpretation downstream from WG1. Even in WG2 and other risk assessment reports, it appears that climate science focus on the most likely part of the distributions has led to collapsing the uncertainty around points (e.g., averages) that are inappropriate for formal risk management. That this makes climate change the odd one out in comparison to other risk domains has been pointed out elsewhere, including in references in

the main article in this thread, and in comments by other authors (notably, RC1, SC2 and SC3).

It seems to me that WG1 authors have an opportunity, dare one say responsibility, to help downstream users in the interpretation of the scientific uncertainty through practical demonstration of how to do this correctly. PPHIS could be a useful tool to turn this principle into practice. This could work alongside (not in opposition to) better collaboration between the different WGs in the IPCC.

**References** Sanborn, A. N., Chater, N. (2016). Bayesian brains without probabilities. Trends in cognitive sciences, 20(12), 883-893.
* * *

---

## Referee Comment (RC2) · F. Zwiers (Referee) · 13 Jul 2018

I recommend acceptance subject to revision, not because I agree with the proposal, but rather because I think the discussion is useful. The article should be revised to provide a more balanced view of the applicability of the uncertainty language, and to recognize that the IPCC has a formal scoping process that produces a scoping report that is approved by the IPCC Plenary and gives scientists direction for the assessment that they should produce. That process could be used to direct the IPCC to produce assessments of high impact scenarios or storylines if this is judged to be desirable by the governments that comprise the IPCC.

[Figure]

Comment:

A potential decision to focus the work of IPCC WGI more heavily on high impact scenarios, or storylines, has little to do with how the IPCC uncertainty language is defined. Rather, this is a scoping issue – that is, one that should be dealt with through the scoping process. If governments feel that this is key information that is required, then they, of course, could seek advice from their scientific communities about whether such an assessment is feasible, and request an assessment if the received advice pointed in that direction. Such an assessment could be included as part of either the full assessment report or, given sufficient literature and importance, could be undertaken as a special report. A key question that arises almost immediately when a high impact scenario or storyline is described is, what are the odds of the occurrence of such an event? It would be entirely reasonable for governments and decision makers to ask scientists to assess, if possible, a likely range for the odds of occurrence of a scenario based on an assessment of the confidence that we have in our understanding of the physical, ecological and socio-economic processes that would produce the scenario. That is, the uncertainty language can be applied to the assessment of high impact storylines just as it can be applied to changes in mean conditions. It is not the uncertainty language that prevents such an analysis of rare, high impact events and the processes that might produce them.

---

## Author Comment (AC3) · 18 Jul 2018

Physically Plausible High Impact Scenarios (PPHIS) for use by IPCC WG1: practical implementation and examples

This comment is an elaboration of my response to the helpful comments by S. Hallegatte, and others, and provides a more detailed explanation of how the proposal could be implemented within the framework of the existing IPCC uncertainty guidelines and associated calibrated language. Several examples are given.

PPHIS Proposed Definition for WG1: An assessed physically based storyline for spe-

cific aspects of future climate change that is consistent with all available evidence and would result in impacts that are substantially greater than those implied by the relevant likely range.

This information is policy relevant because policy makers are concerned with the management of risk.

[revised manuscript text omitted]

---

## Short Comment (SC7) · 20 Jul 2018

Enclosed please find one PDF document and one Excel document. The PDF document is meant to be the written review of the author's idea. The Excel document presents all data used in the composition of the written Word document. These two documents are provided in a zip file titled Risk.zip.

Please also note the supplement to this comment:
https://www.earth-syst-dynam-discuss.net/esd-2018-36/esd-2018-36-SC7-supplement.zip

---

## Short Comment (SC8) · 21 Jul 2018

We would like to address one further point as illustrated in the author's first reference "Climate change a risk assessment." On page 19:

"2. Identify the biggest risks. This follows logically from the first principle. The more a risk could affect our objectives, the more relevance it is likely to have for our decision-making. If risk is defined simply as the product of likelihood and impact, then the biggest risks may be those which are most likely to occur, or those which would have the greatest impact, or those which fall somewhere in between. Mathematically speaking, this will depend on the shape of the probability distribution function. In practice,

the risks of most concern are usually those with the greatest impact, especially when there is potential for irreversible consequences (e.g. death)."

We now believe that the author has confused the words "probability distribution function" (or PDF) with what is really meant which is "cumulative distribution function" (or CDF). We say this because on page 51 (Figure 2), page 52 (Figure 3), pages 58-61 (new chapter Figures 1 and 2) we see the likelihood function on the y-axis going between 0% and 100%, in other words, the likelihood function, we now believe, was always meant to be some form of CDF or probability of exceedance. In fact, throughout this reference we always see the y-axis in either percent or usually an increasing scale of some quantity.

Is anyone here really paying attention to the very fundamentals of risk? How could this happen? I am embarrassed for the author and Professor Ed Hawkins (acknowledgments). As an idealized CDF is rather monotonic and the impact is assumed to be monotonic the resulting R = LI itself will also be monotonic.

We do not claim any subject matter expert (SME) status in risk or impacts (well except for 30+ years as a research coastal engineer where we deal with this stuff all the time), but at least we are able to read a graph with a y-axis labeled "Likelihood" and said "Likelihood" is indeed a CDF.

References King, D., Schrag, D., Dadi, Z., Qui, Y., Ghosh, A. Climate change: a risk assessement. Cambridge University Centre for Science and Policy, Cambridge; 2015.

---

## Short Comment (SC9) · 21 Jul 2018

ISO Guide 73:2009(en) Risk management — Vocabulary

"https://www.iso.org/obp/ui/#iso:std:iso:guide:73:ed-1:v1:en"

5.6 Terms relating to risk analysis

3.6.1.1 likelihood chance of something happening Note 1 to entry: In risk management terminology, the word "likelihood" is used to refer to the chance of something happening, whether defined, measured or determined objectively or subjectively, qualitatively or quantitatively, and described using general terms or mathematically [such as a prob-

ability (3.6.1.4) or a frequency (3.6.1.5) over a given time period]. Note 2 to entry: The English term "likelihood" does not have a direct equivalent in some languages; instead, the equivalent of the term "probability" is often used. However, in English, "probability" is often narrowly interpreted as a mathematical term. Therefore, in risk management terminology, "likelihood" is used with the intent that it should have the same broad interpretation as the term "probability" has in many languages other than English.

3.6.1.4 probability measure of the chance of occurrence expressed as a number between 0 and 1, where 0 is impossibility and 1 is absolute certainty Note 1 to entry: See definition 3.6.1.1, Note 2.

---

## Short Comment (SC10) · 21 Jul 2018

OK, so no more spreadsheets (hopefully). We have now produced the correct risk graph using the ISO standard definition(s) for likelihood/probability. We take the original CDF's and invert them all (instead of R=L (the CDF)*Impact we use R = (1 − CDF)*impact). That is the ISO standard so go argue with the ISO.

The risk plots still exhibit weird asymptotic behaviors due to combining an exponential growth with CDF distributions that have different exponential-like decay behaviors. But things are much clearer now. Contrary to what the author implies, an ECS of 6 degrees centigrade does not create a higher risk than what the IPCC AR5 WGI have specified

in their three conditionals (well, that is, if you are NOT on a fishing expedition and use exponentials combined with poor distribution asymptotic behaviors, in which case you can create all sorts of purported climate fictions).

That is all.

[Figure]

[Figure]

Actual Risk (Using Correct Likelihood Function (1 - CDF))

**Fig. 1.** Risk done right.

---

## Short Comment (SC11) · 23 Jul 2018

We wish to thank the author and the editors of ECD for giving us the opportunity of open review and the ability to submit open public comments. We also wish to thank Dr. Ken Rice (blog online identity ATTP) of the climate science blog And Then There's Physics (aka ATTP);

https://andthentheresphysics.wordpress.com/2018/06/11/low-probability-high-impact-outcomes/

It was this blog that highlighted this discussion paper and allowed us to delve into the

subject matter more completely. The author is encouraged to visit said website to either respond or review or correct the record in that (or any) particular blog discussion.

At this time, we also wish to sincerely apologize to the author(s), Dr. Sutton and (by proxy) Dr. Hawkins, personally and the editors at ESD for any and all untoward and inappropriate statements we made in comments SC7 through to SC10 and any similar dialog in the discussion of this paper that may have occurred at the blog ATTP. There is simply no excusing our behaviors either in the ECD comments section or at the blog ATTP. Thank you again for this opportunity to apologize for our behaviors in the written record. The editors of ECD are, of course, allowed to redact any or all of our comments, in whole or in part, per the ECD public comments guidelines.

---

## Author Comment (AC4) · 31 Jul 2018

I thank Piers Forster for commenting on my proposal. I'm glad we agree that WGI can do a lot better in supporting risk framing than in past reports, but I'm unclear from his remarks what he thinks WG1 should actually do differently, or whether he supports my specific proposal. I certainly do not believe WGI should "go it alone": I proposed that "WGI authors should agree a modest number of key parameters ... [including]... some large-scale impact-relevant metrics (informed by WGII), such as the magnitude of increases in extreme rainfall."

I agree that interaction with WGII on these matters is very important; moreover, devel-

opment of physically plausible high impact scenarios (PPHIS) would provide a helpful new focus for the assessment of climate risks in a way that draws more fully on WGI expertise than in the past. WGII could make use of these scenarios, e.g. to further assess their impacts, and this would be helpful for the Synthesis Report. In my response to the referee comments by Stephane Hallegatte I have provided more explanation of PPHIS, with specific examples of how the concept could be readily applied within the framework of the existing IPCC calibrated language. This more detailed proposal is also attached here. It makes clear that I agree that seeking to quantify tail-likelihood too precisely is unwise, but also that this is not necessary to provide information that is very useful for risk assessment.

There is one point on which I strongly disagree. The comment that the bottom (low impact) tail is "just as important" as the upper (high impact) tail of the uncertainty distribution suggests the reviewer may have forgotten who the audience is. The IPCC is an assessment for policy makers not for climate scientists. For policy makers, high impact scenarios are much more important than low impact ones which means that the IPCC - including WGI - has a duty to provide as detailed an assessment as possible of high impact scenarios. I would also refer Piers to the comments on my proposal by Kris de Meyer and Ted Shepherd.

---

## Author Comment (AC5) · 31 Jul 2018

I fully agree with these comments. In particular, I share the concern that if WGI does not more effectively address the key issues of concern to policy makers than its relevance will diminish, potentially quite rapidly. I also agree that the common misconception that a "conservative" approach to assessing risks is the only one consistent with scientific rigour is a significant barrier to making improvements to the assessment process.

---

## Author Comment (AC6) · 31 Jul 2018

I welcome these thoughtful comments, particularly from a reviewer outside the WGI climate science community. It is very valuable to hear a wider perspective, especially as the subject is the IPCC assessment process whose primary audience is policy makers and not climate scientists. The context from risk assessment practice in other fields is a very important one for climate scientists to appreciate, and I agree with the reviewer that WGI has a responsibility in AR6 to engage much more effectively with the needs of risk assessment.

[Figure]

2018.

---

## Author Comment (AC7) · 31 Jul 2018

I thank E. Sargent for his comments on my proposal. However, I fear he has missed the point of the schematic Figure 1. Its sole purpose in the paper is to provide a visual motivation for why paying specific attention to low likelihood high impact scenarios is a sensible thing to do from the perspective of risk assessment. One might argue that this is such an obvious truth that no illustration is needed to make the point, but it has not been standard practice in IPCC Working Group 1. For this reason, colleagues have found the Figure 1 illustration helpful. I note also that one comment on my proposal (SC5) suggested that the bottom (low impact) tail is "just as important" as the upper

(high impact) tail, which makes little sense from the perspective of risk assessment.

I am accused of a "naïve premise that risk would equal the likelihood times the impact". But this is not a premise I adopt. There is no single universally accepted metric of risk, but it certainly is the case that likelihood x impact is a commonly used measure, as is stated in the paper (and also in King et al, 2015). This means it is a relevant quantity to consider.

It is stated clearly in the caption to Figure 1 that it is a "schematic representation" and also that "The precise shape of the risk curve is dependent on assumptions about the shape of the likelihood and damage functions at high sensitivity (Weitzmann, 2011)". The reviewer appears to have overlooked these key statements.

With regard to whether one should examine a PDF or CDF, there is no simple right or wrong answer, it depends on what the user wants to know. If the user wants to know the risk associated with a specific parameter (such as ECS) exceeding some threshold value then some form of cumulative (integrated) risk estimate may well be useful. However, they may instead be interested to estimate the risk associated with a specific value of this parameter, in which case Figure 1 is an appropriate schematic representation. I note that many of the examples referred to in the King et al report plot the likelihood of exceeding some temperature threshold as a function of time in a warming climate. The likelihood of a fixed but unknown parameter such as ECS having a high or low value is an entirely different class of assessment.

I appreciate that my proposal may be confusing to someone not familiar with the procedures of the Intergovernmental Panel on Climate Change. These procedures were developed in the past for good reasons, but I believe they need to evolve to address more effectively low likelihood high impact scenarios. Further details of my proposal are provided in my response to the referee comments by S. Hallegatte and are attached again here.

---

## Author Comment (AC8) · 31 Jul 2018

I thank Alan Gadian for his comment on my proposal, and I'm glad we agree that IPCC WG1 has a responsibility to pay much greater attention to low likelihood high impact scenarios. I thank him also for pointing me to Kerry Emanuel's highly relevant 2014 comment "Tail Risk vs Alarmism" (http://climatechangenationalforum.org/tail-risk-vs-alarmism/), the substance of which I entirely agree with.

Alan goes beyond the scope of my proposal to criticise IPCC and WG1 climate scientists for failing to include adequately in the assessment process the expertise of the "few . . . who understand the tail", a group which implicitly includes himself. In these

comments he should be wary of hubris. The fact is that no-one fully understands the tail risk because there are deep uncertainties that we cannot fully quantify with current tools and knowledge (see referee comments by S. Hallegatte). These risks are not a product of the meso/micro-scale alone – on the contrary they are influenced by all scales up to that of the Earth as a whole. For this reason global climate models are one important tool amongst others for studying the problem. Like other tools and approaches they have strengths and weaknesses, and must be handled with care.

Tail risks are difficult to assess, but we agree they cannot be ignored and must be afforded greater attention. We also agree that in making the relevant assessments the IPCC must draw on the widest possible range of expertise and literature. In my response to the referee comments by S. Hallegatte I have provided further explanation of how the necessary assessments could be readily included within existing IPCC procedures. I attach this discussion again here.

---

## Author Comment (AC9) · 1 Aug 2018

I thank Francis Zwiers for his comments on my proposal. I'm glad he agrees the discussion is useful. As I understand it he has two specific concerns, to which I respond to here:

1. The IPCC uncertainty language does not preclude the assessment of high impact storylines. I entirely agree with this point, and in my response to referee comments by S. Hallegatte I have provided a more detailed account (also attached here), illustrated with examples, of how the assessment of physically plausible high impact scenarios could be readily handled within the framework of the existing calibrated language. The

point I wish to make is not that the uncertainty language is flawed but rather that by itself it is insufficient to guide the assessment of risk. I am amending the paper to ensure this distinction is spelt out more clearly. The more detailed description, with examples, of how the proposal could be applied will also be included in the final version of the paper.

2. It should be for the IPCC scoping process to decide whether specific attention to low likelihood high impact scenarios is required. Here, I think we do not entirely agree. I of course recognise that the IPCC has a formal scoping process (indeed I was a participant in one stage of this process for AR6, the scoping meeting in Addis Ababa). However, I believe there are weaknesses in this process especially for the main Assessment Report Cycle which require attention and - in view of the importance of the issues and the long-timescales of an IPCC Assessment Cycle - I consider this need urgent. Here is one illustration: since the report is for governments, with the Synthesis Report the headline product, an outsider would naturally expect the scoping process to begin with scoping the Synthesis Report by asking what the IPCC sponsor governments consider to be the most important questions they would like to be assessed. But for historical reasons this is not how it works: the scoping process is much more bottom up, with the scientists making a proposal of what they should assess in each Working Group and the governments merely modifying this proposal in modest ways. Scoping of the Synthesis Report comes only later, whereas in a sensible process the outcomes from scoping the Synthesis Report should be cascaded to the Working Groups so that they receive clear guidance on which issues are of most concern to the governments. Of course there must always be space for scientists to raise issues and concerns that governments may not have thought of, but this does not mean that a fundamentally bottom-up scoping process is appropriate any longer.

A second point about the scoping process is that chapter author teams retain significant latitude (albeit within the constraints of the outline) to decide what questions they will assess. But not surprisingly, the influence of previous practice is very great. Thus

because the approach in five WG1 Assessment Reports has been to assess the likely range and, in general, say very little if anything about the high impact tail, this practice will tend to continue unless there is a very strong steer from the top that it should change.

I would argue that the lack of attention in WGI to low likelihood high impact scenarios is substantially a consequence of weaknesses in the IPCC scoping process. Of course there are other factors, such as the fear of alarmism, which I mentioned in my proposal and was addressed more directly by Kerry Emanuel in his 2014 comment "Tail risk vs. Alarmism" (http://climatechangenationalforum.org/tail-risk-vs-alarmism/#comments). He ended is comment by asking: "Do we not have a professional obligation to talk about the whole probability distribution, given the tough consequences at the tail of the distribution?" I think we do, and that the IPCC WGI should therefore give greater attention to the high impact tail. My Idea presents a simple way in which this could be done, within the constraints of the existing IPCC procedures and the agreed outline of AR6.

It is beyond the scope of my Idea to discuss the IPCC scoping process in any detail. However it is clearly relevant, so if the Editor agrees I would propose to add some brief comments about its relevance to the final version of my paper.

---

## Author Response (AR1)

**Ideas: a simple proposal to improve the contribution of IPCC WG1 to the assessment and communication of climate change risks**

**Responses to Editor Comments**

Editor comments:

I am happy to accept this contribution to the ESD Ideas. I would suggest the author to make minor changes in his submissions as follow:
1) amend the introduction given that the WG1 LA1 meeting has happened now.

*Done*

2) as suggested by Hallegate (and already discussed in the author's response), may be not put to much weight on likelihood as probably very hard to quantify.

*This is made explicit in the new subsection "Practical Implementation and Examples" (see response to Comment 4 below).*

3) as suggested by Forster, not to forget "the bottom part of the pdf". I sympathize with the authors seeing higher than expected impact as more important, but one could argue that a much lower than expected climate response, even if "highly unlikely", could have profound impact on the global economy, energy production, etc. It would be up to the IPCC authors/delegates to decide which of these high risk/low probability event should be discussed and potentially mentioned in the SPM.

*I'm unclear whether there is a disagreement here, or what the Editor is really suggesting. Nowhere do I say "ignore the bottom part of the pdf", and I certainly agree it would be up to the relevant IPCC author teams to decide which PPHIS scenarios should be discussed – albeit with some necessary coordination across WGI, and with input from WGII. I believe this is made clear in my text.*

*Regarding the specific example the Editor gives, this would involve a hypothetical damage function that is not monotonic in (e.g.) ECS, presumably because it is proposed that the world has already invested excessively in expensive mitigation and adaptation measures that would now be wasted. This is conceivable in principle, but I'm unaware of any (balanced) literature that supports such a view.*

4) I see that the author also wrote an additional short note on Physically Plausible High Impact Scenarios (PPHIS). I would suggest to include some of this material (such as the 3 examples) in the original submission.

*I have incorporated this text into a new subsection "Practical Implementation and Examples".*

[revised manuscript text omitted]